# Hypoxia: The Cornerstone of Glioblastoma

**DOI:** 10.3390/ijms222212608

**Published:** 2021-11-22

**Authors:** Marta Domènech, Ainhoa Hernández, Andrea Plaja, Eva Martínez-Balibrea, Carmen Balañà

**Affiliations:** 1B·ARGO (Badalona Applied Research Group of Oncology) Medical Oncology Department, Catalan Institute of Oncology Badalona, 08916 Badalona, Spain; mdomenechv@iconcologia.net (M.D.); ahernandezg@iconcologia.net (A.H.); aplaja.germanstrias@gencat.cat (A.P.); 2Germans Trias i Pujol Research Institute (IGTP), ProCURE Program, Catalan Institute of Oncology, 08916 Badalona, Spain; embalibrea@iconcologia.net

**Keywords:** glioblastoma, hypoxia, HIF-1, HIF-1 inhibitors

## Abstract

Glioblastoma is the most aggressive form of brain tumor in adults and is characterized by the presence of hypervascularization and necrosis, both caused by a hypoxic microenvironment. In this review, we highlight that hypoxia-induced factor 1 (HIF-1), the main factor activated by hypoxia, is an important driver of tumor progression in GB patients. HIF-1α is a transcription factor regulated by the presence or absence of O_2_. The expression of HIF-1 has been related to high-grade gliomas and aggressive tumor behavior. HIF-1 promotes tumor progression via the activation of angiogenesis, immunosuppression, and metabolic reprogramming, promoting cell invasion and survival. Moreover, in GB, HIF-1 is not solely modulated by oxygen but also by oncogenic signaling pathways, such as MAPK/ERK, p53, and PI3K/PTEN. Therefore, the inhibition of the hypoxia pathway could represent an important treatment alternative in a disease with very few therapy options. Here, we review the roles of HIF-1 in GB progression and the inhibitors that have been studied thus far, with the aim of shedding light on this devastating disease.

## 1. Introduction

Glioblastoma (GB) is the most aggressive primary brain tumor. Currently, the standard treatment for patients with GB consists of surgery followed by chemoradiotherapy, which confers a median overall survival rate of around 15 months [1]. Even though the molecular features and the glioblastoma microenvironment have been characterized in depth, no emerging systemic treatment has been found to improve overall survival.

Despite being treated as one single disease, like other tumors, GB has a heterogeneous molecular profile, conferring different biological behaviors. Molecular factors, such as O6-methylguanine DNA methyltransferase (MGMT) methylation status or isocitrate dehydrogenase (IDH) mutations have been shown to be prognostic and predictive biomarkers related to alkylating agents [2] and a prognostic biomarker stronger than histological grade [3], respectively. In fact, IDH mutation confers such a different prognostic that these tumors have been renamed as astrocytoma, IDH-mutant, and grade 4 [4]. As of 2016, molecular analysis is strongly recommended in the case of young patients, in order to specify the diagnosis of GB [5].

Nowadays, a definitive diagnosis of GB requires a pathological examination, which confirms vascular proliferation or necrosis, both features being signs of the presence of tumor hypoxia [6,7]. 

Although GB is one of the most highly vascularized human tumors, this microcirculation is very inefficient, leading to highly hypoxic areas within the tumor and associated necrosis. Hypoxia results in an inadequate nutrient supply and promotes genetic changes; therefore, it is a fundamental aspect in driving tumor growth. Thus, hypoxia-inducible factor 1 (HIF-1), as the master regulator that orchestrates cellular responses to hypoxia, plays a key role in GB aggressiveness [8]. In the context of hypoxia, the expression of angiogenic factors, such as VEGF, EPO and PDGF, is enhanced. In addition, the upregulation of the expression of several factors modulates glucose and fatty acid metabolism, a tumor-immune microenvironment, and autophagy in tumor cells. This review seeks to explore the role of hypoxia in GB patients and the current strategies for inhibiting this pathway.

## 2. What Is the Role of HIF-1? 

HIF-1 was discovered in 1991 as a molecular target associated with intratumoral hypoxia by Semenza et al. [9]. HIF-1 is a transcription factor composed of 2 subunits, α and β. These two subunits have a similar conformation, with a basic helix-loop-helix and a Per/ARNT/Sim (PAS) domain, but their transcription is differentially regulated. The presence of subunit β, also named aryl hydrocarbon receptor nuclear translocator (ARNT), is not oxygen-dependent. On the other hand, the degradation of the α subunit is tightly regulated by oxygen. HIF-1α is stable under hypoxic conditions but is rapidly degraded when oxygen is present. When HIF-1α is stable, it translocates to the nucleus and heterodimerizes with HIF-1β. After dimerization, and once HIF-1 gets into the nucleus, it can induce the transcription of more than one hundred target genes. Physiologically, HIF-1 activates the transcription of a high number of genes that are involved in angiogenesis, cell invasion, autophagy, and glucose and lipid metabolism [10,11,12]. 

In addition, there are two related homologs of HIF-1α, HIF-2α and HIF-3α. HIF-2α, also called endothelial PAS domain protein, has a semi-identical conformation to HIF-1α. HIF -2α is also regulated by the presence or the absence of oxygen, and its degradation is also activated in normoxic conditions, like HIF-1α. Therefore, under hypoxia, it has the ability to bind to HIF-1β and activate the transcription of hypoxia-responsive genes [13,14]. Nevertheless, while HIF-1α is expressed in all cell types, HIF-2α is only expressed in some cell populations, such as endothelial cells and the interstitial cells of some organs. Moreover, although they exert similar actions in a hypoxic environment and share target genes, some of the latter are different [15].

On the other hand, the HIF-3α protein has been shown to trigger a different transcriptional program than its two homologs, HIF-1α and HIF-2α. Importantly, HIF-3α has been shown to negatively regulate HIF-1/2α functions [12,16].

A key aspect of HIF-1α regulation is the presence of the oxygen-dependent degradation domain (ODDD). These ODDDs contain two proline residues that are specifically hydroxylated by prolyl hydroxylase domain (PHD) enzymes under normoxic conditions. 

As a result of the hydroxylation of these domains, HIF-1α is recognized by the Von Hippel–Lindau (VHL) protein. VHL is part of an E3 ubiquitin ligase complex comprising elongin B, elongin C, cullin 2, and RBX1, which leads to the specific degradation of the HIF-1α protein, via the proteasome pathway, in the presence of oxygen [12,17].

In contrast, under hypoxia, PHD enzymes are inhibited; consequently, HIF-1α is not degraded and is stabilized in the cytosol. Afterward, HIF-1α translocates to the nucleus and attaches to the HIF-1β subunit. In addition, HIF-1 requires some cofactors, such as p300/CREB-binding protein (CBP), to be able to bind to the hypoxia-response element (HRE) and to then coordinate the transcriptional response to hypoxia (Figure 1). 

It is worth noting that, apart from the VHL complex, there are other systems regulating HIF-1, such as factor-inhibiting HIF (FIH), which is able to block the interaction between HIF-1 and the transcriptional activators p300/CBP via the asparagine hydroxylation of HIF-1, attenuating HIF transactivation activity [7].

Once HIF-1 reaches the DNA, a high number of genes are specifically activated. It is well known that HIF-1 plays an important role in angiogenesis via the transcription of several genes that are involved in establishing an abnormal vessel network. An important pathway regulated by HIF-1 is that of metabolic reprogramming. HIF-1 enhances the generation and accumulation of lactate via the activation of anaerobic and aerobic glycolysis, glutaminolysis, and modulating lipid metabolism to promote tumor growth.

Moreover, HIF-1 plays a major role in autophagy activation in cancer cells, promoting cell survival and treatment resistance. Finally, hypoxia confers an immunosuppressive tumor microenvironment. Nowadays, this aspect is fundamental since immunotherapy has become the primary focus of interest in cancer treatment.

## 3. HIF-1α Promotes GB Aggressiveness

Hypoxia plays a central role in the GB patient’s prognosis [18]. In particular, HIF-1 is inherently related to GB pathophysiology and has pleiotropic effects on angiogenesis, immunosuppression, cell invasion, and cell survival under hypoxic conditions [11,19] (Figure 1).

Angiogenesis, which is the development of new capillaries from pre-existing blood vessels, plays an essential role in tumor aggressiveness due to its association with the invasion and migration of tumor cells. 

Under hypoxia, HIF-1α controls the expression of several proangiogenic genes, such as vascular endothelial growth factor (VEGF), placenta-like growth factor (PlGF), platelet-derived growth factor (PDGF), angiopoietin (Ang)-1 and -2; erythropoietin (EPO) and insulin-like growth factor 2 (IGF2), which participate in the complex process of angiogenesis [11,20,21,22]. 

Neovascularization is a multistep process. First, tumor cells gather around pre-existing vessels and proliferate after vascular regression occurs, enhancing tumor necrosis; finally, the formation of a new vessel network is established [23].

To accomplish this process, proangiogenic factors bind to their receptors in the endothelial cells, resulting in the degradation and remodeling of the endothelial cell basement membrane and extracellular matrix (ECM). Finally, proangiogenic factors induce the synthesis of a mature vascular basement membrane. These factors stimulate endothelial cell proliferation, assembling into a tubular structure to form stable, new vessels. The formation of these vessel networks is physiologically required for an appropriate nutrient and oxygen supply [24].

VEGF-A is considered the principal protein that promotes the formation of new vessels through the binding of its receptors, mainly VEGFR-2. Indeed, the VEGF family is composed of 5 proteins, VEGF-A, VEGF-B, VEGF-C, VEGF-D, and PIlG.

In fact, VEGF is required for normal blood vessel formation but when VEGF is overexpressed, as in tumor cells, this leads to high vascularity and higher vascular permeability. HIF-1 is considered the principal regulator of VEGF, although VEGF-dependent angiogenesis can also be activated by other factors [25].

In a tumor scenario, these new abnormal vessels help to provide the tumor cells with an adequate oxygen and nutrient supply and facilitate cancer-cell dissemination. On the other hand, these new vessels often have disrupted endothelial junctions due to a lack of pericytes and smooth muscle cell stabilization, which confers higher permeability, resulting in treatment resistance due to lower drug delivery into the tumor.

HIF-1 also promotes angiogenesis via the upregulation of the PDGF proteins. The PDGF family comprises four heparin-binding polypeptide growth factors (A, B, C, and D). PDGF, via the binding and activation of its two cell-surface tyrosine kinase receptors, PDGFR-α and PDGFR-β, stimulates angiogenesis, promoting vessel maturation, pericyte recruitment and VEGF induction. In addition, PDGF activates the oncogenic signaling pathways, such as phosphatidylinositol 3-kinase/protein kinase B (PI3K/AKT) and mitogen-activated protein kinase/extracellular signal-regulated kinase (MAPK/RAS), promoting tumor aggressiveness. 

Moreover, EPO is a hormone that stimulates erythropoiesis (red blood cell production) but, at the same time, it participates in angiogenesis; therefore, it helps to supply nutrition to tumor cells. HIF-1 is the principal regulator of EPO expression; in fact, HIF-1 was discovered due to EPO research [26]. 

Although GB is a hypervascularized tumor, the abnormal vessels do not allow an appropriate oxygen pressure to reach most areas of the tumor, leading to necrosis. While we would expect a low oxygen supply to halt tumor progression, tumor cells can adapt and become more aggressive. In GB, it has been found that tumor cell proliferation is likely to be increased next to anoxic zones [6,18]. 

As mentioned above, hypoxia triggers the formation of microvascular hyperplasia, which, in turn, is an exacerbated form of angiogenesis that occurs in response to the secretion of proangiogenic factors by the cells that form the pseudopalisades [27]. Pseudopalisades, which are hypercellular zones that surround a necrotic area, are a typical feature of GB histology. Furthermore, angiogenesis affords characteristic magnetic resonance images in GB, while neovascularization confers contrast enhancement on the T1 sequence, and tumor-associated edema is translated to a hyperintense image on T2-fluid-attenuated inversion recovery (T2-FLAIR) [28].

Under hypoxic conditions, cancer cells need to adapt through metabolic reprogramming to ensure energy production and cell survival. It is well known that energy production in most cancer cells is via aerobic glycolysis, converting glucose to lactate even when sufficient oxygen is available; this is known as the Warburg effect. Aerobic glycolysis promotes the abnormal proliferation of cancer cells, facilitating malignant progression. HIF-1 plays an essential role in enhancing this glycolytic phenotype in tumor cells [29,30,31,32]. 

HIF-1 promotes the switch of glucose metabolism from mitochondrial oxidative phosphorylation into glycolysis-activating genes, which are involved in extracellular glucose import, and enzymes responsible for the glycolytic breakdown of intracellular glucose. 

During hypoxia, pyruvate is not used by mitochondria; instead, it is converted to lactate in the cytosol due to the upregulation of glycolytic enzymes, inhibiting the conversion of pyruvate to acetyl-CoA via the upregulation of pyruvate dehydrogenase kinase 1 (PDK1).

During aerobic glycolysis, tumor cells only generate two ATP molecules by oxygen-independent glycolysis, instead of the 38 ATP molecules generated by mitochondrial oxidative phosphorylation. Therefore, to assure energy production, HIF-1 facilitates glucose transport into the cytosol, activating the transcription of glucose transporter 1 (GLUT1) and glucose transporter 3 (GLUT3).

Aerobic glycolysis protects tumor cells against cell damage from reactive oxygen species (ROS) and promotes cell survival. To maintain redox homeostasis, tumor cells need to decrease oxidative metabolism to limit the production of ROS [33]. 

HIF-1 promotes lactate production and excretion, activating lactate dehydrogenase A (LDH-A) and monocarboxylate transporter 4 (MCT4) transcription. Extracellular lactate accumulation contributes to the acidification of the microenvironment; both acidosis and lactate indirectly help in the stabilization of HIF-1α [34]. 

Consequently, lactate accumulation promotes angiogenesis, cell motility, cell invasion, radioresistance, immunosuppression and stem cell phenotype promotion.

HIF-1 also contributes to glutamine uptake, which is the second substrate required for cancer cell growth. Glutamine provides the nitrogen required for the synthesis of nucleotides and nonessential amino acids. HIF-1 also enhances glutamine transport into the cancer cell and glutaminolysis. Glutaminolysis contributes to lactate production; initially, glutamine converts to glutamate and α-ketoglutarate and then into malate in the Krebs cycle, finally being converted to lactate [25]. 

Apart from reprogramming glucose metabolism, recent data indicate that many aspects of lipid metabolism are also modified by HIF-1 under hypoxic conditions.

Fatty acids are required for mitochondrial oxidation, energy production, membrane synthesis and energy storage, which make them essential for cell survival.

During hypoxia, there is an increase in fatty acids via the transcription factors: peroxisome proliferator-activated receptors (PPARγ) and fatty acid-binding proteins (FABP) 3 and 7, all activated via HIF-1.

HIF-1 also promotes fatty acid synthesis by a reduction in glutamine metabolism, enhancing the production of α-ketoglutarate, which is converted to citrate, required for the lipid’s synthesis. Moreover, HIF-1 promotes lipid storage in lipid droplets, to avoid the accumulation of the intracellular fatty acids that provoke lipotoxicity [35,36].

HIF-1 induces autophagy to promote tumor cell survival. Autophagy is a catabolic process that is physiologically activated under conditions of oxidative stress or nutrient starvation to recycle metabolic precursors, such as fatty acids, amino acids and ATP; it also activates the degradation of damaged proteins and organelles.

It seems that autophagy has a dual function in cancer, depending on the tumorigenesis stage; at an early stage, it promotes cell survival as a proapoptotic mechanism to prevent tumor initiation but in later stages, it acts as a recycling system for metabolite precursors, contributing to cell survival and tumor growth.

HIF-1 under hypoxia has been shown to activate transcription of the Bcl-2 nineteen-kilodalton interacting protein 3 *(BNIP3*) gene that induces autophagy, displacing Beclin1 from the BCL2 apoptosis regulator (Bcl-2) or B-cell lymphoma-extra-large (Bcl-xL), two proapoptotic proteins [37]. Moreover, BNIP3 that is activated under hypoxic conditions has been shown to play a major role in mitophagy, which protects cancer cells from ROS, promoting cancer-cell survival [38,39,40]. 

Autophagy is also a mechanism of resistance to anti-angiogenic treatment in GB patients. An increase in BNIP3 immunoreactivity was detected in GB patients that progressed after bevacizumab treatment, reflecting higher autophagy activity enhanced by hypoxia [41].

In addition, HIF-1 promotes cancer cell migration and invasion due to the enhancement of epithelial–mesenchymal transition, ECM remodeling and vascular permeability. HIF-1 promotes the mesenchymal phenotype by activating the transcription of the Snail family of zinc-finger transcription factors, consisting of Snail1 (Snail), Snail2 (Slug) and zinc finger E-Box-binding homeobox 1 (ZEB1), which downregulate E-cadherin [39]. HIF-1 facilitates cancer cell invasion, enhancing the disruption of the basement membrane and ECM remodeling. HIF-1 also activates the transcription of several matrix metalloproteinases (MMP-2,9,14), cathepsins, fibronectin and keratins 14, 18 and 19; these degrade components of the ECM and orchestrate ECM remodeling. ECM degradation and the enhancement of the tumor cells to a mesenchymal phenotype favor tumor cell motility [42].

Hypoxia also modulates the tumor microenvironment, favoring immunosuppression, which confers tumor progression and has a poor prognosis. It has been shown to modulate the innate and adaptative immune system.

Initially, GB was thought to be an immune-privileged organ due to the presence of the blood–brain barrier [43]. Afterward, several immune cell types, such as regulatory T cells and M2 polarized macrophages, were discovered in GB, demonstrating that it is one of the “coldest” solid tumors [44,45,46]. GB sustains an immunosuppressive microenvironment promoted by hypoxia via HIF-1 activation, which provokes acidification of the environment, adenosine accumulation and lactate production [47]. 

Tumor-associated macrophages (TAM) are differentiated to M2 polarization, suppressing the host immune system via secreted cytokines favored by hypoxia. HIF-1 also upregulates CD47 expression, which inhibits macrophage phagocytosis [48].

On the other hand, natural killer cells (NK) and lymphocyte T (LT) function are diminished due to the induction of autophagy, blocking the pro-apoptotic signal [49]. HIF-1 enhances CD39/CD73 expression, which provokes an accumulation of adenosine, subsequently blocking LT activity [50]. Moreover, HIF-1 regulates T cells by the induction of aerobic glycolysis in Tregs, which promotes Treg migration, suppressing tumor immunity, while oxidative phosphorylation promotes Treg immunosuppression [31]. Additionally, HIF-1 has been shown to directly upregulate the expression of the programmed death-ligand 1 (PDL-1) in cancer cells [51,52,53].

Angiogenesis plays a central role in tumor immunosuppression. VEGF has been shown to diminish antigen-presenting cells and to induce Treg and myeloid-derived suppressor cells (MDSC). MDSC reciprocally promotes VEGF expression. The transition from M1 to M2 macrophage phenotypes is promoted by VEGF expression [53,54,55,56]. Angiogenesis also creates a barrier that decreases T cell tumor infiltration [57]. In the context of angiogenesis and immunosuppression, Ang-2 is also essential. Ang-2 promotes M2 phenotype macrophage conversion and, through IL-10, promotes the presence of Tregs [58].

Chemoresistance and radioresistance, which favor tumor progression, are also modulated by the lack of oxygen in the tumor. It is well known that radiation therapy is less effective under hypoxic conditions. Ionizing radiation, as an antitumor treatment, consists of directly damaging DNA molecules and the generation of free radicals such as ROS. The reduced oxygen level causes treatment resistance, due to oxygen being required to stabilize the DNA strand breaks caused by radiotherapy [59,60,61,62]. HIF-1 promotes an antioxidant system attenuating ROS production through the activation of PDK1; thus, the decrease of ROS levels provokes radioresistance. 

HIF-1 and, more importantly, HIF-2 are required for the maintenance of cancer stem cells (CSCs) that are capable of self-renewal, and for generating both CSCs and differentiated cells [63]. HIF-1 promotes the CSC phenotype, enhancing the expression of KLF4, MYC, OCT4, SOX2, and NANOG [64,65,66,67]. Glioblastoma stem cells (GSC) are also favored by the stabilization of the Notch intracellular domain by HIF-1α. Moreover, Notch signaling via the activation of epithelial-mesenchymal transition promotes cancer cell invasion. The enhancing stem-cell-like phenotype promotes GB intratumoral heterogeneity and confers both chemo- and radioresistance [68,69].

However, chemoresistance is also favored by HIF-1 activation through VEGF overexpression, promoting the formation of abnormal vessels, which, in turn, restricts antitumoral drugs from reaching the tumor. Additionally, HIF-1 promotes autophagy via BNIP3 transcription, which has been shown to confer cytoprotection.

Finally, Tang et al. showed that the inhibition of HIF-1α, via HIF-1α knocked-down cell models, sensitizes glioma cells to temozolomide, showing a decrease in MGMT expression. As mentioned above, it has also been demonstrated that the benefit of temozolomide, in terms of overall survival, is higher in patients with lower MGMT activity, like patients with MGMT methylation [70,71]. 

Thus, HIF-1 inhibition could reverse GB chemoresistance.

## 4. Is HIF-1 the Main Promotor of Aggressiveness in Brain Tumors?

### 4.1. HIF-1 Expression Is Related to the Brain Tumor’s Grade

HIF-1α has been proposed to drive glioma progression from low-grade astrocytoma to GB; both protein and mRNA expression have been related to a higher pathological tumor grade and poor prognosis. 

It is clear that HIF-1 expression is positively related to VEGF, GLUT-1, GLUT-3, PDGF-C, carbonic anhydrase 9 (CA-9), and osteopontin expression in GB tissue and in GB cell lines [72,73,74].

HIF-1, VEGF, the delta-like canonical Notch ligand 4 (DDL4), and PDGF-C are related to higher microvessel density in both astrocytomas and GB [75,76].

HIF-1 and its targeted genes are related to a higher pathological grade in brain tumors, tumor progression, and treatment resistance; therefore, their expression confers a worse prognosis [19,77,78,79,80,81]. A meta-analysis encompassing 24 studies established that HIF-1 can be related to a higher tumor grade and poorer overall survival with GB [82].

Some of these studies have highlighted that other hypoxia-related markers, such as CA9 and osteopontin, are more robust and could better correlate with the hypoxic microenvironment and aggressiveness of GB than HIF-1. 

Studies in GB tissue showed that HIF-1α immunoreactivity is particularly strong in areas surrounding necrosis that is predominantly in the nuclei, while in low-grade gliomas, HIF-1α expression is predominantly located in the cytosol [81,83].

Protein expression of HIF-1β was also seen to be highly expressed in high-grade tumors; however, it was more widely expressed than HIF-1α, including expression in cells distant from necrotic areas [84].

Moreover, HIF-1 expression is also related to a higher pathological grade in brain tumors other than astrocytoma. Two studies included cases of meningiomas and GB and reported no differences between these two tumor subtypes; however, the expression of HIF-1α was related to tumor progression in both meningiomas and astrocytic tumors, showing a greater presence of HIF-1α in high-grade tumors compared with low-grade tumors [85,86]. In oligodendroglioma patients, HIF-1 protein expression in tissue was also associated with a worse prognosis [87].

### 4.2. HIF-1α Participates in Signaling Pathways Involved in GB

Apart from the presence or the absence of oxygen, HIF-1 is regulated by several mechanisms. An important regulation in tumors is via the activation of oncogenes, such as EGFR (epidermal growth factor receptor), or the loss of tumor suppressors, such as p53 or phosphatase and the tensin homolog (PTEN) [6].

HIF-1α is well known as a downstream gene of the oncogenic pathway PI3K/AKT/mammalian target of rapamycin (mTOR). PI3K upregulates HIF-1α protein synthesis via AKT/mTOR and the eukaryotic translation initiation factor 4E (eIF-4E) binding protein (4E-BP1) Nevertheless, the activation of AKT could also lead to increased HIF-1α expression in tumor cells via another pathway that is independent of mTOR activation [88,89,90,91,92,93]. The loss of PTEN has also been shown to enhance HIF-1 transcription via the activation of the PI3K/AKT/mTOR pathway. In fact, the Cancer Genome Atlas (TCGA) reported 41% of deletion mutations in PTEN and 25% of PI3K-activating mutations in GB databases [94].

The amplification of EGFR is present in more than 50% of GB patients and is one of the most common molecular characteristics. Thus, in GB, HIF-1α transcription can also be activated via the MAPK/ERK signaling pathway/signal transducer and activator of transcription 3 (STAT3) [30]. 

SRC is a proto-oncogene non-receptor tyrosine kinase that leads to the activation of several downstream pathways, such as PI3K/AKT, MAPK/ERK and STAT3. The SRC protein is hyperphosphorylated in GB cells, especially when they are under hypoxia conditions. SRC hyperactivation is achieved through the binding of its domains, SH2 and SH3, to the cytoplasmic portions of activated receptor tyrosine kinases (RTK) (such as EGFR, PDGFR or VEGFR); however, no amplification or activation mutations have been detected. SRC promotes GB invasiveness due to the upregulation of MMP-2, 9 and fibroblasts, via the TGFβ pathway; it also promotes the overexpression of proinflammatory transcription factors. Despite being activated due to hypoxia, SRC has been shown to suffer hyperphosphorylation, due to ionizing radiation conferring radioresistance via enhancing cancer cell adhesion, motility, survival, and proliferation [95]. Therefore, preclinical studies blocking SRC are being undertaken as a strategy to radiosensitize GB cells [96].

The p53 tumor suppressor pathway is also implicated in the degradation of HIF-1 in GB patients. The loss of p53, which is detected in approximately 30% of GB patients, has been associated with elevated levels of HIF-1α under normoxic conditions. HIF-1α binds to p53, allowing MDM2 (mouse double minute 2 homolog) mediated ubiquitination and the proteasomal degradation of HIF-1α. Therefore, the loss of, or mutations in, p53 revoke the MDM2-mediated degradation of HIF-1α (Figure 1) [97,98].

Finally, IDH mutant GB presents higher HIF-1α levels than wild-type tumors. IDH-1 mutations result in a decrease in cellular α-ketoglutarate, which is required by PHD enzymes that hydroxylate, and promote the degradation of HIF-1α via VHL, resulting in a higher stabilization of HIF-1α [99,100].

## 5. Can We Block HIF-1 to Stop Tumor Progression?

Considering the relevance of HIF-1 in GB, its inhibition has been proposed as a potential strategy to treat patients. Inhibiting the HIF-1 pathway would impact tumor dedifferentiation, angiogenesis, and autophagy, reducing the development of cytotoxic resistance and improving patient survival.

Bevacizumab, a monoclonal antibody targeting VEGF-A, is one of the main downstream genes of HIF-1, preventing its interaction with the VEGF receptor tyrosine kinases. In May 2009, an accelerated approval was granted for bevacizumab treatment of recurrent GB in those patients whose prior treatment failed, combined with another drug. Bevacizumab in monotherapy has been approved by the Food and Drug Administration, but its approval was refused by the European Medicines Agency due to a lack of demonstrated benefit regarding overall survival in clinical trials [101,102].

Mendez et al. showed that the migration capacity of GB cells is impaired in in vitro models, with both human and mouse glioma cells, due to HIF-1α activation. Their in vitro experiments with GB cell lines showed that under hypoxic conditions, migration was significantly higher compared with control cell cultures under normoxic conditions. Furthermore, they showed that under normoxic conditions, knocked-down HIF-1α glioma cells had a lower migration capacity than control cells [103]. 

Although many approaches have been tried to inhibit HIF-1, drugs that only target specific components of the hypoxia signaling pathway have generally failed to produce an enduring clinical response. On the other hand, there is a lack of biomarkers to validate HIF-1 inhibition in pre-clinical models and tumor tissue. One should also consider that in GB, only drugs penetrating the blood–brain barrier (BBB) should be taken into account [9]. In this context, several drugs have been explored; a few of them have been tested in early-stage clinical trials (Table 1). Some of these are discussed in the following section.

## 6. Downregulating HIF-1 mRNA Expression

Aminoflavone (AFP464), a ligand of the aryl hydrocarbon receptor (AhR) that dimerizes with HIF-1α, has been shown to downregulate HIF-1 mRNA. Although the exact mechanism by which HIF-1 mRNA is decreased has yet to be elucidated, it has been proposed that it may be due to the modulation of HIF-1 transcription. A phase I clinical trial (NCT01015521) in breast cancer patients reported serious adverse events, such as thrombosis, hepatitis and pulmonary toxicity, and no tumor responses were observed [104,105]. 

EZN-2698 is an antisense oligonucleotide that binds specifically and with high affinity to HIF-1 mRNA in in vitro and in vivo models. The binding has been shown to provoke the downregulation of mRNA levels and a reduction of the HIF-1 protein. Treatment with EZN-2968 showed downregulation of HIF-1 target genes, such as VEGF and MMP-2, in both prostate cancer and GB cell models. In addition, it showed a decrease in tumor cell proliferation and a recession in tube formation of endothelial cells. The results of a phase I clinical trial (NCT01120288) in patients with refractory advanced solid tumors showed that EZN-2968 has a good safety profile. This could demonstrate the downregulation of HIF-1 expression in tumor biopsies and confirm preliminary antitumoral activity. Nevertheless, the drug development was suspended [106,107]. Recently a phase Ib (NCT01672463) trial with OKN-007 in GB recurrent patients was shown to be safe and to have an overall survival of 21 months [108]. In mice models, OKN-007 reduces HIF-1, GLUT-1 and VEGFR2 expression under hypoxia conditions. Moreover, this small molecule was seen to cross the BBB and to halt pro-apoptotic and anti-angiogenic activities [109].

## 7. Inhibiting HIF-1 at the Protein Level

Some small molecules targeting HIF-1 have been identified by screening tumor cells with natural chemical libraries. As a result of the screening, two compounds, KC7F2 and 103D5R, were reported to decrease HIF-1 levels by inhibiting its translation [110,111,112]. 

As mentioned above, HIF-1α is highly regulated by multiple signaling pathways; drugs inhibiting these pathways have been shown to modulate HIF-1 protein translation. In this sense, several agents, already approved for cancer treatment, affect the rate of HIF-1 protein synthesis, including receptor tyrosine kinase inhibitors, cyclin-dependent kinase inhibitors, p53 activators, and also other approved therapies like the inhibitors of topoisomerase I and II, such as topotecan, EZN-2208, and microtubule-disrupting agents [113,114,115]. Cardiac glycosides are used as potential anti-cancer agents due to their effects on the inhibition of proliferation and induction of apoptosis and/or autophagy in cancer cells, they have been shown to inhibit the HIF-1 protein, but not to downregulate HIF-1 expression [116,117]. Nevertheless, they do downregulate VEGF, diminish neutrosphere formation, and decrease CD133 expression, which is a marker of GSC.

## 8. Promoting HIF-1 Degradation

A third form of inhibition consists of modulating HIF stability. Some inhibitors do so by inducing its degradation. 

Vorinostat (SAHA), a histone/protein deacetylase (HDAC) inhibitor, is a potent class of tumor-suppressive agents. It exerts its effect by loosening tightly wound chromatin and alters the hypoxia pathway via the inhibition of HIF-1α nuclear translocation. It has also been proven to downregulate transcriptional activity, showing the downregulation of HIF-1 target genes, such as endothelin 1, EPO, gut 1, and VEGF. SAHA acetylates its associated chaperone, shock protein 90 (Hsp90), which controls its protein folding. However, Hsp90 participates in the protein-folding of a large number of proteins, so it is difficult to determine whether the cause of the antitumor activity is truly related to HIF-1 inhibition [118]. The same mechanism of action has been demonstrated for IDF-1174, which inhibits HSP70 chaperone activity and also suppresses the accumulation of HIF-1α [119].

LBH589 (panobinostat), a hydroxamic acid, demonstrated its potent inhibitory activity against all class I HDAC, inducing HIF-1α inhibition at the protein level under hypoxic conditions. This agent is currently undergoing several trials in combination with other drugs in different cancer types; however, panobinostat failed to demonstrate activity in monotherapy in melanoma patients and was associated with high hematological toxicity (NCT01065467) [120,121]. 

A small molecule, PX-478, has shown potent antitumor activity in mouse models, that seems to be associated with HIF-1α levels within the tumor. It was proven to be safe in a phase I clinical trial (NCT00522652) in patients with advanced solid tumors, with a prolonged stable disease as the best response [122,123]. PX-478 inhibits HIF-1 at the protein level, inhibiting HIF-1 deubiquitination, thus favoring the presence of polyubiquitinated HIF-1 and, consequently, showing the downregulation of VEGF expression.

Icaritin is a product of the Chinese herb genus, *Epimedium*, and has demonstrated antitumoral activity via the inhibition of cell proliferation, the promotion of cell differentiation, and apoptosis induction. Although the exact mechanism of action of icaritin is not well established, in in vitro experiments in GB cell models it significantly inhibited cell invasion by targeting the extracellular matrix MMP via the PTEN/Akt/HIF-1α signaling pathway [124]. Its effects as an immune modulator in tumors are being studied in phase III trials; icaritin has already demonstrated a good safety profile and has achieved preliminary durable responses in advanced hepatocellular carcinoma patients in a phase I clinical trial (NCT02496949).

## 9. Interfering in HIF-1 to HIF-1 HRE Binding 

To promote gene transcription, HIF-1 needs to bind to specific DNA sequences called HREs. Some substances have been shown to interfere with this binding, consequently affecting HIF-1 function.

Echinomycin, a cyclic peptide from the family of quinoxaline antibiotics, is known to bind DNA in a sequence-specific manner and, therefore, to inhibit HIF-1 transcriptional activity. So far, clinical phase I trials have not shown any activity against different solid tumors, but otherwise causing severe nausea and vomiting as adverse events. Anthracyclines are already approved for the treatment of cancer patients, exerting their cytotoxic activity via a number of different mechanisms, among which they have been shown to inhibit HIF-1 transcriptional activity by blocking its binding to HRE sequences [125]. Acriflavine, another drug approved as an antifungal treatment, has also been shown to have inhibitory activity by blocking HIF-1 DNA binding and, in fact, to possess antitumoral activity in in vitro studies using glioma cells [126,127].

The transcriptional activity of HIF-1 can also be diminished by inhibiting its coactivators; for example, chetomin interferes in the interaction between HIF-1 and its co-activator, p300. However, due to its high toxicity profile, the development of chetomin has not been further studied [128].

KCN1 is a synthetic sulfonamide that also interferes in the interaction between HIF-1 and its coactivators p300/CBP. Experimentally, it was shown to specifically inhibit HIF-1 transcriptional activity in glioma cell lines, demonstrating downregulation of HIF-1 downstream genes. Furthermore, KCN1 in GB in mice models was shown to penetrate the blood–brain barrier; therefore, it could be a potential treatment in GB patients [129].

Despite several strategies being tried to block HIF-1 activity, success has not been demonstrated in early clinical trials. Due to the mechanism of action of HIF-1, one might expect that the best way to block activity would be at the post-transcriptional level or at the level of interaction with DNA, but although an effect downregulating downstream genes has been demonstrated, no relevant antitumoral activity has been shown.

**Table 1 ijms-22-12608-t001:** HIF-1inhibitors.

Drug Name	Mechanism of Action	Clinical Trial	Phase	Glioma Models	Ref
EZN-2208	Downregulation of HIF-1 mRNA	NCT01251926	Phase I	Yes	[115]
Aminoflavone	Blockage of HIF-1α dimerization	NCT0101552	Phase I	No	[104,105]
EZN-2698	Downregulation of HIF-1 mRNA	NCT01120288	Phase I	Yes	[106,107]
KC7F2	Inhibition of HIF-1 translation	-	-	Yes	[110,111]
103D5R	Inhibition of HIF-1 translation	-	-	Yes	[110,112]
PX-478	Downregulation of HIF-1 protein level	NCT00522652	Phase I		
LBH589	Downregulation of HIF-1 protein level	NCT01065467 NCT00859222	Phase I Phase II	Yes	[72,73]
Topotecan	Inhibition of HIF-1 protein synthesis	Approved for other cancer types	-	Yes	[114]
OKN-007	Downregulation of HIF-1 mRNA	NCT01672463	Phase Ib	Yes	[108,109]
Cardiac glycosides	Downregulation of HIF-1 protein synthesis	-	-	Yes	[77,78]
Vorinostat	Degradation of the HIF-1 protein	Approved for other cancer treatments	- -	No	[118]
Echinomycin	Blockage of HIF-1 DNA binding	Approved	-	Yes	[82]
Acriflavine	Blockage of HIF-1 DNA binding	Antifungal approved drug	-	Yes	[83,84]
Icaritin	Degradation of HIF-1α	NCT02496949	Phase I	Yes	[124]
KCN1	Blockage of HIF-1 and coactivators p300/CBP	-	-	Yes	[129]
Chetomin	Blockage of HIF-1 and coactivators p300/CBP	-	-		[85]

## 10. Conclusions

The hypoxic microenvironment is the main feature that confers GB its aggressiveness and treatment resistance. HIF-1 is the master regulator of hypoxia that promotes the activation of the transcription of the hundreds of genes involved in several features of tumor progression, such as angiogenesis, autophagy, tumor metabolism, immunosuppression, and cell invasion. 

New treatment strategies tested in recurrent GB patients, such as immunotherapy or targeted therapies to overcome cancer progression, have failed to demonstrate benefits in terms of overall survival [130,131]. Nowadays, immunotherapy has become a focus of interest in cancer treatment; thus, the search for strategies to modulate the immune tumor microenvironment is fundamental. 

For this reason, HIF-1α inhibition represents a hope for halting GB progression and modulating the tumor’s immune microenvironment.

HIF-1 inhibition could enhance the immune GB microenvironment; therefore, the anti-VEGF monoclonal antibody, bevacizumab, in combination with the immune checkpoint inhibitor, nivolumab, was tested in a phase III trial. The Checkmate 143 trial failed to demonstrate a benefit in terms of overall survival in comparison with bevacizumab monotherapy in patients with recurrent GB [131,132,133]. Although encouraging, this treatment showed that the single inhibition of VEGF is not sufficient to reactivate the tumor immune microenvironment to enhance immune checkpoint inhibitor activity. The inhibition of M2 phenotype macrophage recruitment could stand as an immunotherapy strategy, in combination with HIF-1 inhibition in GB patients. 

It is thought that the complete inhibition of HIF-1 is necessary to show potent antitumor activity and to promote the activation of the immune system. Bevacizumab, which solely inhibits VEGF, provokes an important inhibition of angiogenesis that diminishes tumor edema and attenuates neurological symptoms but does not prolong patient survival [22].

So, due to the central role of HIF-1, which participates in multiple pathways that promote aggressive tumor behavior, finding a drug that specifically inhibits HIF-1 could change the GB patient’s treatment paradigm.

Nevertheless, several molecules have demonstrated the ability to inhibit HIF-1 in preclinical studies. HIF-1 inhibition was shown to decrease tumor proliferation, migration, and invasion and, also, to downregulate HIF-1 downstream genes. On the other hand, in clinical trials in patients with refractory solid tumors, very little antitumoral effect has been shown. We would expect that HIF-1 inhibition would lead to a significant benefit for GB patients, due to its central role in the development of this tumor. One of the main reasons for this limited effect could be the lack of specificity of HIF-1α inhibitors. 

This review points out that HIF-1 is involved in GB, promoting tumor aggressiveness, and affecting several hallmarks of cancer; thus, an effort should be made in order to find a strong and specific HIF-1 inhibitor [134]. 

The inhibition of HIF-2α, which can also block the hypoxia pathway, should also be considered an attractive strategy for GB treatment. Recently, HIF-2 has attracted scientific interest, due to HIF-2α overexpression, specifically in glioblastoma cells and GCS, but not in normal tissue [135]. Instead, HIF-1 is expressed in both cell types. Moreover, HIF-2 also plays a central role in cancers under hypoxic conditions, such as GB. For example, the HIF-2 inhibitor, PT2385, presented antitumoral activity in a phase I trial (NCT02293980) in patients with clear renal cell carcinoma, but no responses were demonstrated in a phase II (NCT03216499) trial with recurrent GB patients [136,137]. Nevertheless, other HIF-2α inhibitors that are currently under research could block GB progression.

In addition, the inhibition of HIF-1 downstream genes has been widely studied and showed anti-tumor activity in mice models, However, due to the scant efficacy demonstrated with HIF-1 inhibitors, we consider that a complete blockage of HIF-1, and not solely the inhibition of one of its downstream pathways, should be the focus for further research. 

Under normoxic conditions, the participation of the VHL complex in HIF-1 proteasome degradation is composed of various molecules, some of which proved to be HIF-1-specific and should be further studied. The modulation of the components of this complex promoting HIF-1 degradation could, therefore, specifically inhibit HIF-1, enhancing its degradation even under hypoxic conditions, as, for example, UBXN7 [138,139]. 

Although HIF-1 has a central role in GB, to date no HIF-1-specific inhibitor has been approved for this subset of patients. Other strategies to inhibit this pathway need to be considered and reliable biomarkers that prove HIF-1 inhibition should be explored.

## Figures and Tables

**Figure 1 ijms-22-12608-f001:**
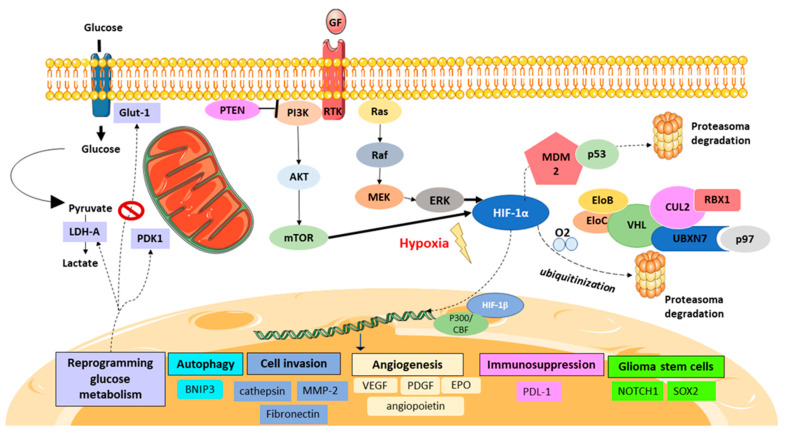
HIF-1 signaling pathway in GB. The presence of oxygen (O_2_) leads HIF-1α ubiquitination and its proteasome degradation via the VHL complex. Hypoxia stabilizes HIF-1α after it translocates to the nucleus, heterodimerizing with HIF-1β, after its binding to DNA, helped by the coactivators P300/CBF to perform its action as a transcription factor. The PI3K/PTEN and MAPK/ERK pathways con also activate HIF-1 due to activator mutations and due to the activation of receptor tyrosine kinase (RTK) or the presence of growth factor (GF). HIF-1 activation promotes the transcription of several genes activating angiogenesis, metabolic reprogramming, cell invasion, immunosuppression, and cancer stem cell phenotypes.

## Data Availability

Data sharing not applicable.

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
