# Peer review of "Hypoxia: The Cornerstone of Glioblastoma"

_ijms, 2021, doi:10.3390/ijms222212608_

Round 1

Reviewer 1 Report

The authors of this manuscript discuss the key role of hypoxia in glioblastoma (GBM) and its association with aggressiveness and pathophysiological mechanisms.  The originality and novelty of the review is good. Most of concepts are fully examined and correctly reported. Indeed, the descriptions of HIF-1 and hypoxia mechanisms in GBM is not limited to the basic notions, reporting several correlations with additional hallmarks of GBM, such as invasion, migration, metabolism reprogramming and immunomodulation.  The authors investigated and highlighted specific topics related to hypoxia, discussing molecular processes and biological pathways. A list of strategies to block hypoxia in GBM are showed in the final part of the manuscript opening new perspectives and offering a good vision of the state of art for the scientific community.

I suggest just three minor corrections:

  1. Line 242: GB is one of the of the “coldest” solid tumors. May the authors clarify the “coldest” meaning?
  2. I suggest citing the following paper “The Role of Hypoxia and SRC Tyrosine Kinase in Glioblastoma Invasiveness and Radioresistance” PMID:33020459, Cancers (Basel). 2020.Here, it is reported how SRC tyrosine protein is linked to PI3K/AKT/mTOR and MAPK/ERK signaling pathways in radioresistance and invasion mediated by hypoxia in GBM. They may also cite "SRC Tyrosine Kinase Inhibitor and X-rays Combined Effect on Glioblastoma Cell Lines". Int J Mol Sci. 2020;21(11):3917. Published 2020 May 30. doi:10.3390/ijms21113917. Here a good strategy combinig a specific SRC inhibitor and radiotherapy in hypoxic GBM cells is showed. 
  3. Resolution and quality of writings and font in Figure 1 might be improved.

Author Response

ID: ijms-1477592
Title: Hypoxia: the cornerstone of glioblastoma
Internation Journal of Molecular Science

Response to Reviewer 1’s comments:

The authors of this manuscript discuss the key role of hypoxia in glioblastoma (GBM) and its association with aggressiveness and pathophysiological mechanisms. The originality and novelty of the review is good. Most of concepts are fully examined and correctly reported. Indeed, the descriptions of HIF-1 and hypoxia mechanisms in GBM is not limited to the basic notions, reporting several correlations with additional hallmarks of GBM, such as invasion, migration, metabolism reprogramming and immunomodulation. The authors investigated and highlighted specific topics related to hypoxia, discussing molecular processes and biological pathways. A list of strategies to block hypoxia in GBM are showed in the final part of the manuscript opening new perspectives and offering a good vision of the state of art for the scientific community.

First of all, we would like to thank the reviewers for their thorough review and excellent comments, which have greatly improved the quality of our manuscript. Please find below our point-by-point replies to the comments:

I suggest just three minor corrections:

1.Line 242: GB is one of the of the “coldest” solid tumors. May the authors clarify the “coldest” meaning?

We really appreciate the reviewer for his/her suggestion. We tried to explain better the meening of “cold” tumor in the context of immune microenvironment.

We added a sentence paragraph in section 3 (page 5) including new information and new references in the current revised version, as follows:

Initially, GB was thought to be an immune privileged organ due to the presence blood brain barrier [43] . Afterwards several immune cell types, such as regulatory T cells and M2 polarizated macrophages, were discovered in GB, demonstrating is one of the of the “coldest” solid tumors [44] [45][46]. GB sustains an immunosuppressive mi-croenvironment promoted by hypoxia, via via HIF-1 activation which which pro-vokes an acidification of the environment, adenosine accumulation and lactate pro-duction [47].

2.I suggest citing the following paper “The Role of Hypoxia and SRC Tyrosine Kinase in Glioblastoma Invasiveness and Radioresistance” PMID:33020459, Cancers (Basel). 2020.Here, it is reported how SRC tyrosine protein is linked to PI3K/AKT/mTOR and MAPK/ERK signaling pathways in radioresistance and invasion mediated by hypoxia in GBM. They may also cite "SRC Tyrosine Kinase Inhibitor and X-rays Combined Effect on Glioblastoma Cell Lines". Int J Mol Sci. 2020;21(11):3917. Published 2020 May 30. doi:10.3390/ijms21113917. Here a good strategy combinig a specific SRC inhibitor and radiotherapy in hypoxic GBM cells is showed.

We thank the reviewer to suggest these two works in order to improve the information related involved pathways and radioresistance.

We added this new information in the current revised version, as follows(pages 8-9):

SRC is proto-oncogene non receptor tyrosine kinase which leads to the activation of several downstream pathways such as PI3K/AKT, MAPK/ERK and STAT3. SRC protein is hyperphosphorylated in GB cells specially when they are under hypoxia conditions. SRC hyperactivation is through binding of its domains, SH2 and SH3, to the cytoplasmic portion of activated receptor tyrosine kinases (RTK) (such as EGFR, PDGFR or VEGFR), instead no amplification or activation mutations have been de-tected. SRC promotes GB invasiveness due to the upregulation of MMP-2, 9 and fibro-blasts via the TGFβ pathway, it also promotes the overexpression of proinflammatory transcription factors. Despite being activated due to hypoxia SRC has shown to suffer an hyperphosphorylation due to ionizing radiation conferring radioresistance via en-hancing cancer cell adhesion, motility, survival and proliferation[96]. Therefore, pre-clinical studies blocking SRC are being undertaken as an strategy to radiosensitize GB cells[97].

3.Resolution and quality of writings and font in Figure 1 might be improved.

We thank the reviewer for pointing this out and we enlarge the font of the writing of Figure 1.

Reviewer 2 Report

The topic of the current review is actual. I like the form of the presented data. My suggestions will reflect the only opinion, what authors may add to improve the general scientific value. 

  1. In Introduction you have well-described glioblastoma as a linkage to HIF. In the title of the review, you have the word “hypoxia”. Not only HIF. It will be better for readers if you will add 2-3 sentences concerning proangiogenic markers or other players important in hypoxia. You do it in the third section, but maybe something should be added in the Introduction.
  2. When we speak about the role of HIF in the tumor microenvironment, in the context of angiogenesis – is a wider situation. Maybe, you can more detailed describe the possible involvement of HIF in angiogenesis and add more effector cells/mechanisms (even a short description) or omit this. I understand your idea and excellent explanation but should give that recommendation.
  3. Line 184 – please delete the bracket.

Author Response

ID: ijms-1477592
Title: Hypoxia: the cornerstone of glioblastoma
Internation Journal of Molecular Science

Response to Reviewer 2’s comments:

The topic of the current review is actual. I like the form of the presented data. My suggestions will reflect the only opinion, what authors may add to improve the general scientific value.

We thank the reviewers for their thorough review and excellent comments, which have greatly improved the quality of our manuscript. We specially really appreciate the constructive comments of the reviewer 2.

Please find below our point-by-point replies to the comments:

In Introduction you have well-described glioblastoma as a linkage to HIF. In the title of the review, you have the word “hypoxia”. Not only HIF. It will be better for readers if you will add 2-3 sentences concerning proangiogenic markers or other players important in hypoxia. You do it in the third section, but maybe something should be added in the Introduction.

We really thank the reviewer 2 for the suggestion to add some information in the introduction.

We have now included this information in the current revised version of the manuscript (current page 1 and 2), as follows:

In the context of hypoxia, the expression of angiogenic factors , such as VEGF, EPO and PDGF, are enhanced. In addition the upregulation of the expression of several factors modulate glucose and fatty acids metabolism, tumor immune microenvironment and autophagy in tumor cells.

2.When we speak about the role of HIF in the tumor microenvironment, in the context of angiogenesis – is a wider situation. Maybe, you can more detailed describe the possible involvement of HIF in angiogenesis and add more effector cells/mechanisms (even a short description) or omit this. I understand your idea and excellent explanation but should give that recommendation.

We thank the reviewer for pointing this out and we added some information of the association of angiogenesis and immunesupression.

We added this information as follows:

Angiogenesis plays a central role in tumor immunosuppression. HIF-1 and VEGF showedwere shown to diminish antigen-presenting cells and to induce regulatory T cells (Treg) and, myeloid-derived suppressor cells (MDSC). MDSC reciprocally promotes VEGF expression. The transition from M1 to, and M2 macrophages phenotype is promoted by VEGF expressionTAM recruitment in GB [54][55][56][57]. Angiogenesis also creates a barrier which decreases T cell tumor infiltration [58]. In the context of angiogenesis and immunosupression Ang-2 is also essential. Ang-2 promotes M2 phenotype macrophages conversion and through IL-10 promotes the presence of T regs[59].Moreover, HIF-1 regulates T cells by the induction of aerobic glycolysis in Tregs, which promotes Treg migration, suppressing tumor immunity, while oxidative phosphorylation promotes Treg immunosuppression [31].

3.Line 184 – please delete the bracket.

We thank the reviewer for detecting this typing mistake and we corrected in the current versions of this revised manuscript.
